# Phylogeography of *Baryancistrus xanthellus* (Siluriformes: Loricariidae), a rheophilic catfish endemic to the Xingu River basin in eastern Amazonia

**Keila Xavier Magalhães**[1], **Raimundo Darley Figueiredo da Silva**[2], **André Oliveira Sawakuchi**[3], **Alany Pedrosa Gonçalves**[4], **Grazielle Fernanda Evangelista Gomes**[2], **Janice Muriel-Cunha**[5], **Mark H. Sabaj**[6], **Leandro Melo de Sousa**[1]*

1 Laboratório de Ictiologia de Altamira, Universidade Federal do Pará, Altamira, Pará, Brazil, 2 Laboratório de Genética Aplicada, Instituto de Estudos Costeiros, Universidade Federal do Pará, Bragança, Pará, Brazil, 3 Instituto de Geociências, Universidade de São Paulo, São Paulo, São Paulo, Brazil, 4 Instituto Nacional de Pesquisa da Amazônia, INPA, Programa de Pós-Graduação em Biologia de Água Doce e Pesca Interior, Manaus, Amazonas, Brazil, 5 Instituto de Estudos Costeiros, Campus Bragança, Universidade Federal do Pará, Bragança, Pará, Brazil, 6 Department of Ichthyology, The Academy of Natural Sciences of Drexel University, Philadelphia, PA, United States of America

* leandro.m.sousa@gmail.com

**Data Availability Statement:** All relevant data are within the manuscript.

## Abstract

*Baryancistrus xanthellus* (Loricariidae) is an endemic fish species from the Xingu River basin with its life history in the shallow rapid waters flowing over bedrock substrates. In order to investigate the genetic diversity and demographic history of *B. xanthellus* we analyzed sequence data for one mitochondrial gene (Cyt b) and introns 1 and 5 of nuclear genes Prolactin (Prl) and Ribosomal Protein L3 (RPL3). The analyses contain 358 specimens of *B. xanthellus* from 39 localities distributed throughout its range. The number of genetically diverged groups was estimated using Bayesian inference on Cyt b haplotypes. Haplotype networks, AMOVA and pairwise fixation index was used to evaluate population structure and gene flow. Historical demography was inferred through neutrality tests and the Extended Bayesian Skyline Plot (EBSP) method. Five longitudinally distributed Cyt b haplogroups for *B. xanthellus* were identified in the Xingu River and its major tributaries, the Bacajá and Iriri. The demographic analysis suggests that rapids habitats have expanded in the Iriri and Lower Xingu rivers since 200 ka (thousand years) ago. This expansion is possibly related to an increase in water discharge as a consequence of higher rainfall across eastern Amazonia. Conversely, this climate shift also would have promoted zones of sediment trapping and reduction of rocky habitats in the Xingu River channel upstream of the Iriri River mouth. Populations of *B. xanthellus* showed strong genetic structure along the free-flowing river channels of the Xingu and its major tributaries, the Bacajá and Iriri. The recent impoundment of the Middle Xingu channel for the Belo Monte hydroelectric dam may isolate populations at the downstream limit of the species distribution. Therefore, future conservation plans must consider the genetic diversity of *B. xanthellus* throughout its range.

**Funding:** 1. KXM and APG received master and doctoral fellowship, Coordenação de Aperfeiçoamento de Pessoal de Nível Superior - Brasil (CAPES Finance Code 001). 2. AOS and LMS receive grants from CNPq (304727/2017-2 and 309815/2017–7 respectively). 3. AOS received FAPESP grants (2012/50260-6 and 2016/02656-9). 4. LMS received grant from CNPq (Edital Universal, proc. 486376/2013-3). 5. JM-C received a grant from FAPESPA/VALE (043/2011). 6. Research supported in part by iXingu Project (NSF DEB–1257813, MHS).

**Competing interests:** The authors have declared that no competing interests exist.

## Introduction

Loricariidae (sucker-mouth armored catfishes) is easily the most diverse family of catfishes (Siluriformes) with approximately five [1] or six [2] subfamilies, 115 genera and over 1000 species widely distributed throughout the freshwaters of South and Central America [3, 4]. The largest loricariid subfamily, Hypostominae, contains about 45 genera and 491 species [3, 4]. Although species of Hypostominae are morphologically conserved, pigmentation patterns are often highly variable and have been used to distinguish numerous forms at and below the species level. Such variation is often geographically tied to a specific river basin, or sometimes longitudinal stretches within a river basin (e.g., [5]).

Large clearwater river systems draining shield terrains in the Amazon and Orinoco basins hold the highest diversity of Hypostominae. Such rivers have naturally transparent waters due to low suspended sediment loads [6, 7]. The three largest clearwater rivers in the Amazon Basin, the Tocantins, Xingu and Tapajós, respectively, support the largest faunas of Hypostominae (e.g., [8]). Those three basins largely drain the Brazilian Shield uplands. Only their downstream portions lie on the sedimentary terrains of the lowland Amazon, where each river's channel becomes naturally flooded and forms an estuary-like channel named as "fluvial ria" [9]. Among those three clearwater tributaries, the Xingu River stands out due to the extremely complex and unique channel morphology when flowing over the fractured bedrocks of the Brazilian Shield. Known as Volta Grande do Xingu, this broadly zig-zag sinuous stretch is divided into a network of bedrock channels with numerous rocky rapids [10, 11].

The genus *Baryancistrus* includes eight described species with three occurring in the Xingu Basin, the endemics *Baryancistrus chrysolomus* and *Baryancistrus xanthellus* and the more widely distributed *Baryancistrus niveatus*. *Baryancistrus xanthellus* is restricted to where the channels of the Xingu River and its major tributaries, the Iriri and Bacajá, flow over Precambrian igneous and metamorphic rocks [12] of the Brazilian Shield (Fig 1). Commonly known as golden-nugget pleco, this species is recognized by having gold-colored spots on the body and conspicuous yellow bands along the distal margins of the caudal and dorsal fins [13]. Because of its attractive color pattern, this species is highly popular as an ornamental fish [14]. In certain parts of the Xingu Basin, local communities consume *B. xanthellus* due to its abundance and relatively large adult size (up to 300 mm SL).

*Baryancistrus xanthellus* is usually found in rapids over rocky substrates that vary in composition and heterogeneity. Specimens have been collected at depths greater than 20 m in flows of moderate to strong velocity. Like most loricariids, *B. xanthellus* is a nocturnal fish and classified as detritivore, feeding mostly on algae and aufwuchs [13, 15].

It is observed a large variation in pigmentation between populations of *B. xanthellus* from distinct localities in the Xingu, Iriri and Bacajá rivers. Several colour patterns are distinguished based on the relative size, intensity and distribution of yellow spots and distal bands on dorsal and caudal fins (Fig 1).

This remarkable polychromatism within *B. xanthellus* may be influenced by local changes in environmental conditions. The role of environmental changes in the diversification of Amazonian biota is critical to understanding the origin of immense tropical biodiversity [16]. Most studies on the genetic diversity of Amazonian biota have focused on forest taxa (e.g., [17–20]) with relatively few studies targeting aquatic taxa (e.g., [21]). Studies of population genetics and phylogeography are needed to infer historical events and processes that account for the current-day distribution of lineages and partitioning of genetic diversity within species [22, 23]. Thus, the phylogeographic patterns of *B. xanthellus* are studied here and compared to climate changes that affected habitats of eastern Amazon rivers during the late Quaternary in order to

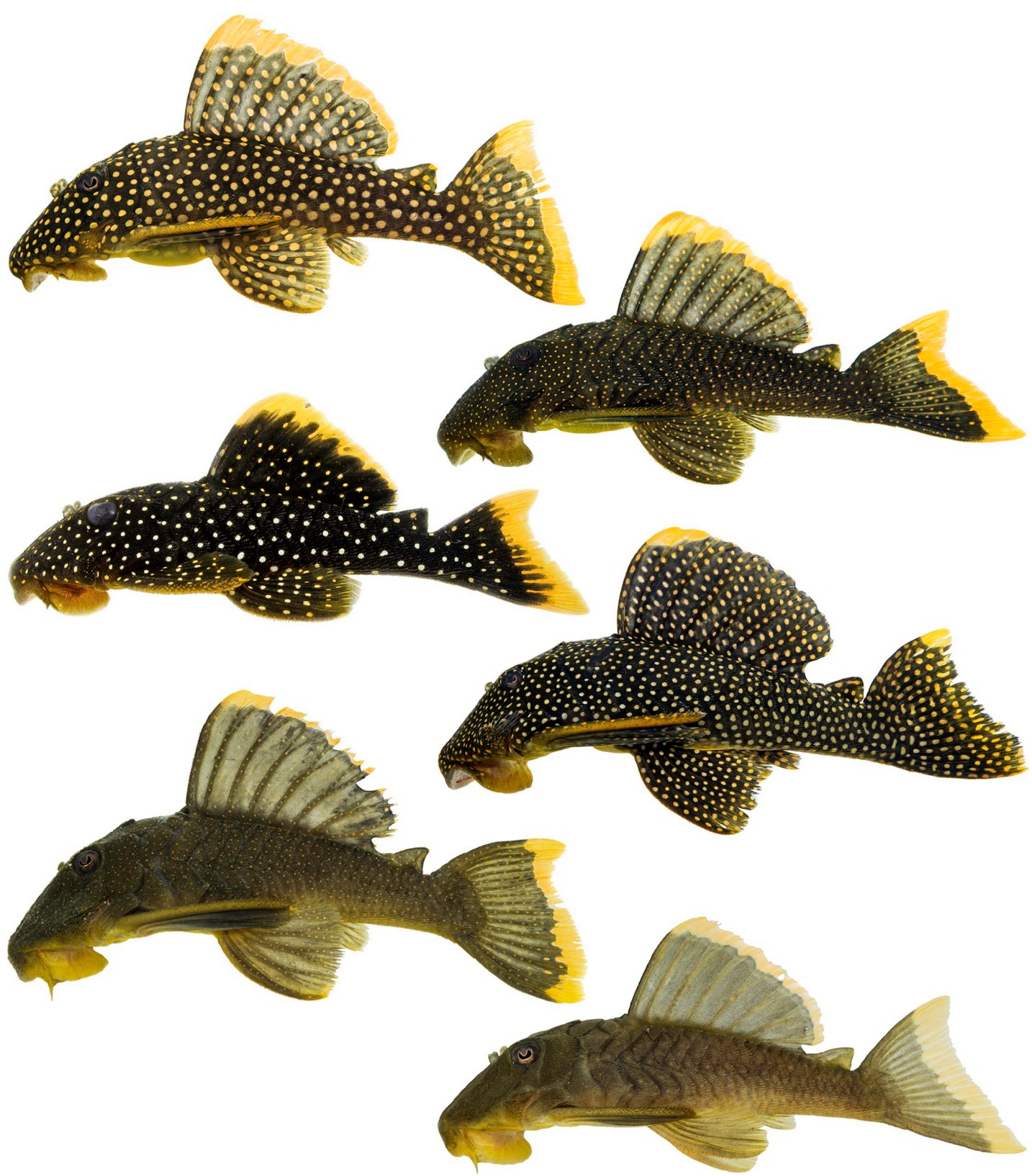

**Fig 1. Different color patterns present in *Baryancistrus xanthellus*.** Photographs of live specimens taken right after collection. Not all color morphs are represented in the figure.

shed light on the processes influencing the diversification of rheophilic fishes in the rapids of large clearwater rivers.

Furthermore, considering its endemicity, commercial value, and importance as a protein source, detailed studies are needed on the population genetics and phylogeography of *B. xanthellus*. Those data can help design proper management plans and effective conservation strategies for this species in the Xingu River and its tributaries. Such studies are further compelled by the recent construction of the Belo Monte Dam Complex in the middle Xingu River [10, 24, 25].

This study evaluates genetic differentiation within *B. xanthellus* based on one mitochondrial gene (Cyt b) and introns 1 and 5 of nuclear genes Prolactin (Prl) and Ribosomal Protein L3 (RPL3), respectively. The main goal of this work is to understand the phylogeography and population structure throughout most of the species' range in the Xingu River basin. Our results are used to infer underlying historical factors and to discuss the conservation of *B. xanthellus* with respect to hydrological and fluvial landscape changes imposed by the Belo Monte Dam Complex. Such changes include the loss of rocky rapids due to the flooding of the Xingu channel above the impoundment dam and the severe dewatering and suppression of the flood pulse below the dam.

## Materials and methods

### Fieldwork

Analyzed specimens were collected at 39 localities in the Xingu Basin, including 24 sites in the main channel from Vitória do Xingu to São Félix do Xingu upstream, 13 sites in the Iriri River, the largest Xingu tributary, and two sites in the Bacajá River, another major tributary that flows into Volta Grande stretch of the middle Xingu (Fig 2 and Table 1). No experimentation was conducted on live specimens. All material was collected in accordance with Brazilian law, under scientific collection licence (SISBIO 31089–1) and was in accordance with the precepts of the ethics committee for animal use in research "Comissão de Ética no Uso de Animais (CEUA)" of Universidade Federal do Pará (UFPA), protocol number 3032120919.

From 2012 to 2016, a total of 358 specimens of *B. xanthellus* were collected from throughout its range in the Xingu (n = 257), Iriri (77) and Bacajá (24) rivers. All specimen identifications were performed in the field and later confirmed using the original published description [13]. A small portion of muscle tissue or pelvic fin were removed from each individual and stored in cryogenic tubes of 96–99% ethanol. Voucher specimens were individually tagged, fixed in 10% formaldehyde, stored in ethanol 70% and cataloged into the zoological collections of the following institutions: Laboratório de Ictiologia de Altamira, Universidade Federal do Pará (LIA-UFPA), Brazil; Fish Collection of the Instituto Nacional de Pesquisas da Amazônia (INPA-ICT), Brazil, and The Academy of Natural Sciences of Philadelphia (ANSP), USA.

### Labwork

Total DNA was isolated using the Wizard Genomic kit (Promega) according to manufacturer instructions. To access the quality of DNA, samples were run on a 1% agarose gel using a mixture of 2 μL of DNA and 2 μL of Gel Red™ (Biotium). After electrophoresis at 60 Volts for 30 minutes, DNA samples were visualized under ultraviolet light and photo-documented.

Three genomic regions were amplified via Polymerase Chain Reaction (PCR), including one mitochondrial gene, cytochrome b (Cyt b) and introns 1 and 5 of nuclear genes Prolactin (PRL) and Ribosomal Protein L3 (RPL3), respectively. Primers and annealing temperatures used for each marker are shown in Table 2.

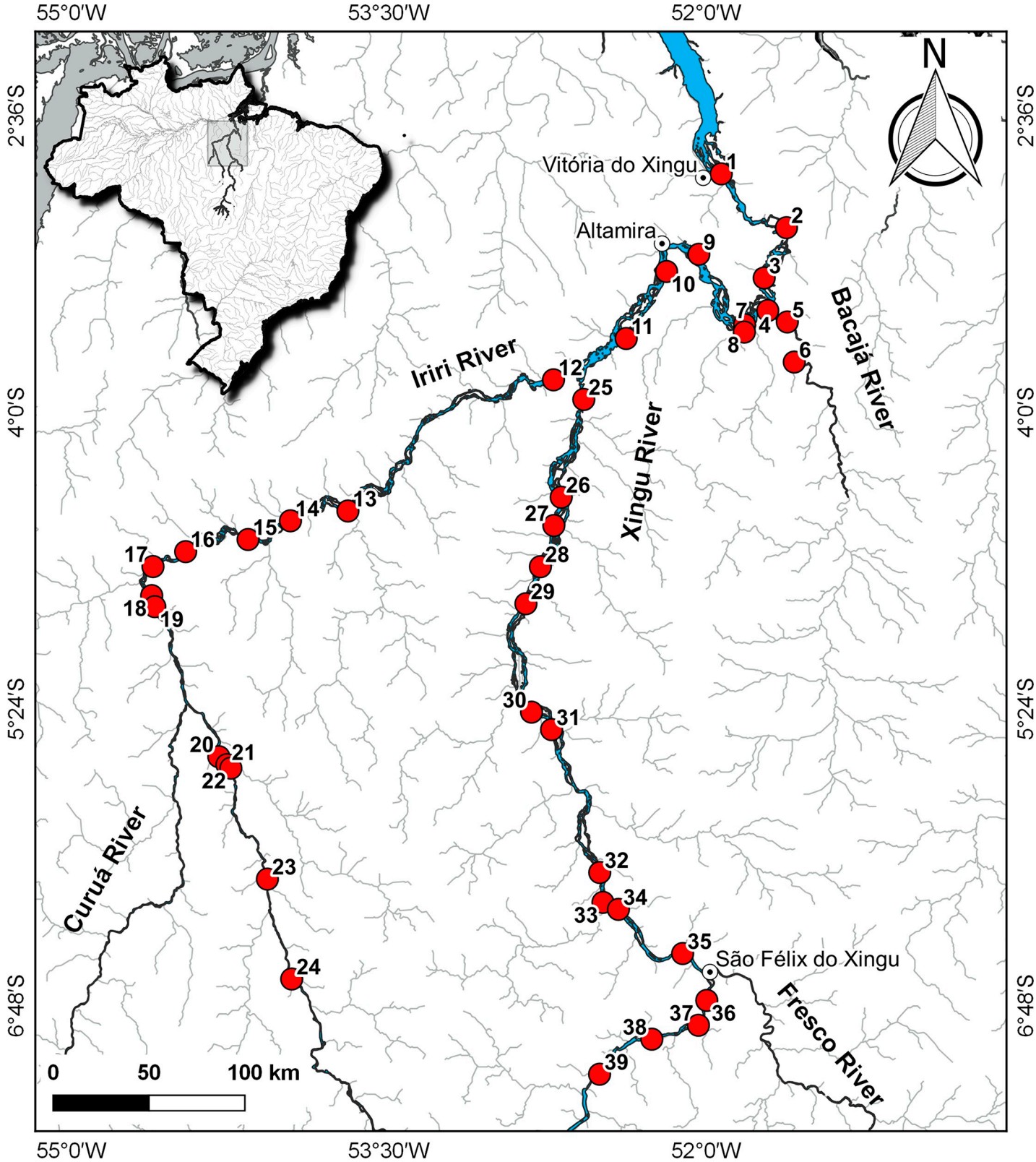

**Fig 2. Map of collection sites of *Baryancistrus xanthellus* in the Xingu River basin; stretch of channel impounded by Belo Monte Dam (between site 8 and 10) not shown.** 1: Vitória do Xingu; 2: Belo Monte; 3: Jericoá; 4: Percata; 5: Pariaxá; 6: Seca Farinha; 7: Ilha da Fazenda; 8: Igarapé Itatá; 9: Cotovelo; 10: Gorgulho da Rita; 11: Boa Esperança; 12: Cachoeira Grande; 13: Boa Esperança (Resex); 14: Cachoeira das Minhocas; 15: Ilha do Papagaio; 16: Cachoerinha; 17: Ressaca da Califórnia; 18:

Lajeiro; 19: São Lucas; 20: Bené; 21: Pousada; 22: Porto Zé Carlos; 23: Zéfa; 24: Irineu; 25: Acima da confluência; 26: Balisa; 27: Pedra Preta (Resex); 28: Estragado; 29: Morro Grande; 30: Bela Vista; 31: Bom Jardim; 32: São Gonçalo; 33: Serra do Pardo; 34: Travessão do Nazaré; 35: Araraquara; 36: Remansinho; 37: Xadai; 38: Pedra Preta; 39: Onça.

**Table 1. Number of individuals of *Baryancistrus xanthellus* sampled at each site.** Sites numbered according to the map in Fig 1; first letter of site abbreviation corresponds to river (i.e., "X" Xingu, "I" Iriri, "B" Bacajá).

| Locality | | No. of individuals |
|---|---|---|
| 1 | Vitória do Xingu (XVX) | 1 |
| 2 | Belo Monte (XBM) | 37 |
| 3 | Jericoá (XJC) | 22 |
| 4 | Percata (BPT) | 1 |
| 5 | Pariaxá (BPX) | 13 |
| 6 | Seca Farinha (BSF) | 10 |
| 7 | Ilha da Fazenda (XIF) | 30 |
| 8 | Igarapé Itatá (XII) | 4 |
| 9 | Cotovelo (XCT) | 23 |
| 10 | Gorgulho da Rita (XGR) | 8 |
| 11 | Boa Esperança (XBE) | 14 |
| 12 | Cachoeira Grande (ICG) | 25 |
| 13 | Boa Esperança/Iriri (IBE) | 1 |
| 14 | Cachoeira das Minhocas (ICL) | 4 |
| 15 | Papagaio Island (IIP) | 5 |
| 16 | Cachoeirinha (ICA) | 5 |
| 17 | Ressaca da Califórnia (IRC) | 1 |
| 18 | Lajeiro (ILJ) | 10 |
| 19 | São Lucas (ISL) | 4 |
| 20 | Bené (IBN) | 6 |
| 21 | Pousada (IPD) | 1 |
| 22 | Porto Zé Carlos (IPC) | 8 |
| 23 | Zefá (IZF) | 6 |
| 24 | Irineu (IIU) | 1 |
| 25 | Acima da Confluência (XCC) | 5 |
| 26 | Balisa (XBL) | 2 |
| 27 | Pedra Preta/Resex (XRP) | 8 |
| 28 | Estragado (XED) | 5 |
| 29 | Morro Grande (XMG) | 1 |
| 30 | Bela Vista (XBV) | 2 |
| 31 | Bom Jardim (XBJ) | 2 |
| 32 | São Gonçalo (XSG) | 5 |
| 33 | Serra do Pardo (XSP) | 5 |
| 34 | Travessão do Nazaré (XTZ) | 2 |
| 35 | Araraquara (XAR) | 16 |
| 36 | Remansinho (XRE) | 5 |
| 37 | Xadai (XXA) | 15 |
| 38 | Pedra Preta (XPP) | 15 |
| 39 | Onça (XON) | 30 |
| | **TOTAL** | **358** |

**Table 2. Primer sequences and PCR annealing temperatures used for each marker in the present study.**

| Marker | Primers | Sequence (5'- 3') | Reference | Annealing (°C) |
|--------|---------|-------------------|-----------|----------------|
| Cyt b | FishCytbF | ACCACCGTTGTTATTCAACTACAAGAAC | [26] | 55 |
| | TrucCytbR | CCGACTTCCGGATTACAAGACCG | | |
| RPL 3 | RPL35F | AAGAAGTCYCACCTCATGGAGAT | [27] | 54 |
| | RPL36R | TTRCGKGGCAGTTTCTTTGTGTGCCA | | |
| Prl | Prl1F | GACAARCTKCACTCBCTCAGCCA | [28] | 64 |
| | Prl1R | TGNAGDGAGGABGTGTGRCAC | | |

PCR products were purified in PEG (Polyethylene Glycol) according to [29] and sequenced using the dideoxy-terminal method [30] with Big Dye kit reagents (ABI Prism™ Dye Termi-nator Cycle Sequencing Reading Reaction) and an ABI 3500 automatic sequencer (Life Technologies, Foster City, CA, USA).

## Data preparation

Sequences were manually edited in BioEdit [31] and aligned using the CLUSTALW method [32] available in that software. In the case of nuclear markers, the algorithm Phase v.2.0 [33] available in DNAsp v.5.10 [34] was applied to score the gametic phases of each individual. Five runs of PHASE consisted of 1,000 burn-in iterations and 1,000 principal iterations, with a thinning interval of 1. Output files from Phase were converted using the software SeqPhase [35]. Only haplotypes with probability values >0.6 were used in further analyses. Recombination events were estimated according to the pairwise homoplasy index (PHI) of [36] available in SplitsTree v.4.14.4 [37].

The number and frequency of haplotypes was evaluated in the software DNAsp v.5.10 [34], which was also used to prepare the input files for the software Arlequin [38].

## Population and phylogeographic analyses

Descriptive parameters for each marker, such as number of polymorphic sites, and nucleotide ($\pi$) and haplotype ($h$) diversities, were established using the software Arlequin v.3.5.1.2 [38]. To verify the spatial and genealogic relationship of haplotypes for each marker, a haplotype network was built in the software Haploviewer [39] based on a Maximum Likelihood tree obtained in the PhyML package [40].

We applied the method implemented in BAPS 6 [41] to assign individuals to genetic clusters. With this method, the number of groups (i.e., genetically diverged lineages) is defined according to a Bayesian algorithm that estimates the distribution of allele frequencies in the tested populations [41]. Each analysis was performed 20 times for every level of K (1–20), with five replicates for each K. The K with the highest posterior probability (HPP) was chosen as the correct clustering.

Analysis of molecular variance (AMOVA) [42] was used to evaluate the genetic differentiation the population groups of *B. xanthellus* indicated by BAPS. AMOVA was carried out in Arlequin v.3.5.1.2 [38]. The significance level was calculated by conducting 10,000 permutations.

A geographic representation of the distribution and frequency of haplotypes was performed using the software GenGIS v2.2.0. [43].

To test for deviations from neutrality due to evolutionary processes within populations of *B. xanthellus*, the *Fs* [44] and *D* [45] values were estimated in Arlequin v.3.5.1.2 [38] based on 10,000 permutations.

Demographic history of *B. xanthellus* was inferred using the Extended Bayesian Skyline Plot (EBSP) method available in the software BEAST v.1.8 [46]. EBSP analysis was based on 100 million generations with genealogies sampled each 10,000 steps, according to the evolutionary model suggested by Kakusan v.4 [47] (i.e., HKY+G). A strict molecular clock was assumed based on a mutation rate of 0.76%/site/million of years within lineages, as suggested by Zardoya and Doadrio [48] for cytochrome b.

We checked convergence by visually inspecting and computing the effective sample size higher than 200) for each parameter in two independent runs using the program Tracer v.1.6 (available at http://beast.bio.ed.ac.uk/tracer), setting the burn-in to 10%. The remaining parameters and priors were used with the default values. EBSPs were plotted using R [49].

# Results

## Dataset and levels of genetic diversity

Sequence data for the 717-bp fragment of Cyt b revealed 169 polymorphic sites and 135 haplotypes among 358 individuals; 69% of those haplotypes were unique. BAPS analysis of the Cyt b fragment clustered individuals of *B. xanthellus* into five genetically diverged groups lettered A through E (Fig 1). The most frequent haplotype (H41) was shared by 69 individuals from Group C, followed by H4, shared by 19 individuals from Group A.

PHI tests assigned non-significant p-values to both introns, RPL3 (p = 0.9059) and Prl (p = 0.1623), indicating an absence of recombination events. For the RPL3 intron, 60 haplotypes were identified in a 225-bp fragment sequenced for 221 individuals. Haplotype H1 was shared by 123 individuals, followed by H4, present in 42 individuals. For the Prl intron, 203 haplotypes were found in a final fragment of 479 bp obtained from 224 individuals. The most frequent haplotypes for the Prl intron were H1 and H5, shared by 59 and 32 individuals, respectively (Table 3).

For the five genetically diverged groups (A–E) inferred from the BAPS analysis, Cyt b haplotype diversity (*h*) ranged from 0.944 ± 0.012 (Group D) to 0.275 ± 0.148 (E) and nucleotide diversity (*π*) varied between 0.0023 ± 0.0016 (B) and 0.0076 ± 0.0041 (D). For the RPL3 nuclear intron, values of haplotype and nucleotide diversity ranged from *h* = 0.809 ± 0.023 (A) to 0.244 ± 0.052 (D) and *π* = 0.0061 ± 0.0042 (A) to 0.0014 ± 0.0016 (D). In the case of Prl nuclear intron, *h* values varied from 0.971 ± 0.006 (C) to 0.894 ± 0.078 (E) while *π* values ranged from 0.0075 ± 0.0043 (B) to 0.0048 ± 0.0030 (D) (Table 3).

## Population genetic structure

Based on the mitochondrial marker (717-bp fragment of Cyt b), the BAPS analysis discriminated five genetically diverged groups within *B. xanthellus* (K = 5; logML = -3688.8922; Posterior Probability = 1) (Fig 3). Group A, named Resex of Iriri River, comprised 71 specimens from 12 sites along the Iriri and three sites in the Xingu channel, one immediately above and the first two sites downstream the Iriri-Xingu confluence. Group B (Resex of Xingu River) included 21 individuals from nine sites in the Xingu channel, all but one distributed from the first two sites downstream the Iriri-Xingu confluence upstream to about the midpoint between that confluence and São Felix do Xingu. Group C (Volta Grande do Xingu) included the largest number of individuals (159) and dominated the downstream-most limit of the species range, specifically the upper Xingu Ria, Middle Xingu channel downstream the mouth of the Iriri, and Bacajá River. Group D (São Félix do Xingu) comprised 92 individuals and dominated the upstream-most sites sampled in the Xingu channel. Group E (Ecological Station of Terra do Meio) included the fewest number of individuals (15) and was restricted to the upstream-most sites sampled in the Iriri River.

**Table 3. Values of genetic diversity and neutrality *Fs* and D tests in *Baryancistrus xanthellus* based on the five genetically diverged groups (A–E) previously identified by BAPS analysis of Cyt b marker (N = number of individuals, Nh = number of haplotypes, S = polymorphic sites, h = haplotype diversity, and π = nucleotide diversity.**

| Population | N | Nh | S | h±dp | π±dp | Tajima (D) | Fu (FS) |
|---|---|---|---|---|---|---|---|
| **Cyt b** | | | | | | | |
| A | 71 | 26 | 28 | 0.891 +/- 0.025 | 0.0033 +/- 0.0020 | -1,875* | -20,025** |
| B | 21 | 12 | 14 | 0.910 +/- 0.048 | 0.0023 +/- 0.0016 | -2,059** | -8,681** |
| C | 159 | 57 | 67 | 0.806 +/- 0.033 | 0.0027 +/- 0.0017 | -2,585** | -27,356** |
| D | 92 | 37 | 45 | 0.944 +/- 0.012 | 0.0076 +/- 0.0041 | -1,219 | -18,205** |
| E | 15 | 3 | 15 | 0.275 +/- 0.148 | 0.0030 +/- 0.0020 | -2,225** | 2,795 |
| **RPL3** | | | | | | | |
| A | 106 | 12 | 9 | 0.809 +/- 0.023 | 0.0061 +/-0.0042 | -0,513 | -4,171* |
| B | 42 | 8 | 7 | 0.453 +/- 0.094 | 0.0031 +/- 0.0027 | -1,563* | -4,922* |
| C | 162 | 25 | 17 | 0.799 +/- 0.026 | 0.0059 +/- 0.0041 | -1,484* | -21,532** |
| D | 122 | 11 | 6 | 0.244 +/- 0.052 | 0.0014 +/- 0.0016 | -1,571* | -13,493** |
| E | 10 | 4 | 4 | 0.533 +/- 0.180 | 0.0054 +/- 0.0043 | -0,521 | -0,459 |
| **PRL** | | | | | | | |
| A | 100 | 57 | 40 | 0.966 +/- 0.009 | 0.0071 +/- 0.0041 | -1,701* | -26,219** |
| B | 42 | 23 | 22 | 0.962 +/- 0.013 | 0.0075 +/- 0.0043 | -0,986 | -14,194** |
| C | 170 | 84 | 34 | 0.971 +/- 0.006 | 0.0053 +/- 0.0032 | -1,675* | -26,839** |
| D | 124 | 31 | 18 | 0.930 +/- 0.010 | 0.0048 +/- 0.0030 | -0,847 | -23,352** |
| E | 12 | 8 | 9 | 0.894 +/- 0.078 | 0.0060 +/- 0.0038 | -0,118 | -2,683 |

**p < 0.01.

*p < 0.05.

The sample number (N) for introns is duplicated due to heterozygosity.

Genetically diverged groups B, C and D of *B. xanthellus* were restricted to the Xingu channel. Their frequency of occurrence formed a longitudinal gradient with the downstream range of the species dominated by Group B individuals, the middle portion dominated by Group C,

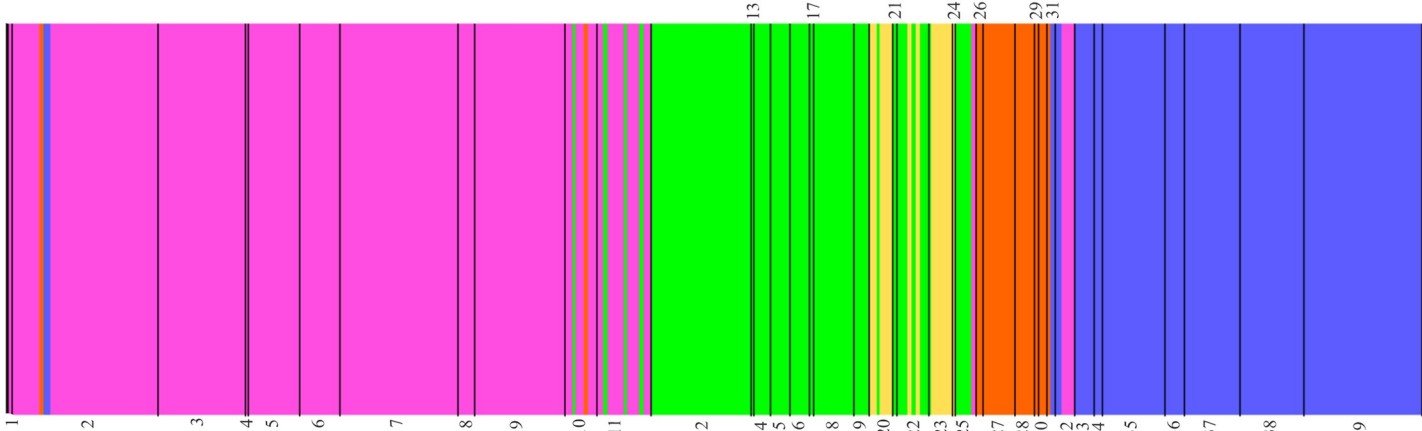

**Fig 3. Estimation of genetically diverged groups in *Baryancistrus xanthellus* in the Xingu River basin inferred from the Bayesian analysis (BAPS) of 717-bp Cyt b fragment in 358 individuals.** The five groups are indicated by green (A), orange (B), magenta (C), blue (D) and yellow (E). The numbers represent the collection sites. 1: Vitória do Xingu; 2: Belo Monte; 3: Jericoá; 4: Percata; 5: Pariaxá; 6: Seca Farinha; 7: Ilha da Fazenda; 8: Igarapé Itatá; 9: Cotovelo; 10: Gorgulho da Rita; 11: Boa Esperança; 12: Cachoeira Grande; 13: Boa Esperança (Resex); 14: Cachoeira das Minhocas; 15: Ilha do Papagaio; 16: Cachoeirinha; 17: Ressaca da Califórnia; 18: Lajeiro; 19: São Lucas; 20: Bené; 21: Pousada; 22: Porto Zé Carlos; 23: Zéfa; 24: Irineu; 25: Acima da confluência; 26: Balisa; 27: Pedra Preta (Resex); 28: Estragado; 29: Morro Grande; 30: Bela Vista; 31: Bom Jardim; 32: São Gonçalo; 33: Serra do Pardo; 34: Travessão do Nazaré; 35: Araraquara; 36: Remansinho; 37: Xadai; 38: Pedra Preta; 39: Onça.

**Table 4. Analysis of Molecular Variance (AMOVA) to evaluate the genetic differentiation among and within the five genetically diverged groups of *B. xanthellus* indicated by BAPS.**

| Source of variation | Variance components | Variation % | Statistic ϕ |
|---|---|---|---|
| **Cyt b** | | | |
| Among Groups | 6.50929 Va | 81.64 | ϕST = 0.816* |
| Within Groups | 1.46341 Vb | 18.36 | |
| **RPL3** | | | |
| Among Groups | 0.05293 Va | 9.62 | ϕST = 0.096* |
| Within Groups | 0.49746 Vb | 90.38 | |
| **PRL** | | | |
| Among Groups | 0.20710 Va | 12.14 | ϕST = 0.12* |
| Within Groups | 1.49900 Vb | 87.86 | |

Asterisks indicate significant values (p < 0.01).

and the upstream range, above São Felix do Xingu, exclusively Group D. Groups A and E were restricted to the Iriri channel except for a few Group A individuals found in the Xingu channel short distances upstream and downstream the mouth of the Iriri. Group E individuals had the smallest distribution and were limited to the furthest upstream sites in the Iriri River.

For Cyt b, the AMOVA (Table 4) showed that most of the genetic variation was found between the five genetically diverged groups (81.64%) instead of within groups (18.36%) (ϕST = 0.816). A different scenario was revealed from the analysis of nuclear regions. For both RPL3 and Prl introns, the AMOVA indicated that most of the variation is found within (90.38% and 87.86%, respectively) instead of among groups (Table 4).

Substructure indices in populations of *B. xanthellus* from the Xingu River basin were corroborated by the allocation of Cyt b haplotypes to five clusters in the haplotype network. Group C comprised most of specimens from the region identified as Volta Grande do Xingu. The other groups discriminate the samples from São Félix do Xingu, Xingu Resex, Iriri Resex and the Ecological Station of Terra do Meio, revealing longitudinally structured populations along Xingu and Iriri rivers (Figs 4 and 5).

The lack of genetic substructure in nuclear markers was evident in the haplotype network, since no correlation was observed between haplotype and geographic distribution (Figs 6 and 7).

## Demographic history

Negative and significant values were obtained in the *Fs* and *D* tests using both mitochondrial and nuclear markers for nearly all five population groups of *B. xanthellus*, indicating deviations from neutrality (Table 3).

The demographic history of each identified group (A–E) was inferred by Extended Bayesian Skyline Plots. The graphs obtained for groups A and C indicated a population expansion initiated 200 ka (thousand years ago). On the other hand, the effective population sizes (Ne) of groups B and D remained stable, while group E experienced a recent population expansion estimated at 5 ka (Fig 8).

## Discussion

### Genetic diversity, population structure and phylogeography

High levels of haplotype diversity (*h*) and low variation in nucleotide diversity (*π*) were observed in Cyt b sequences for three of the five groups analyzed: Iriri Resex (group A), Xingu

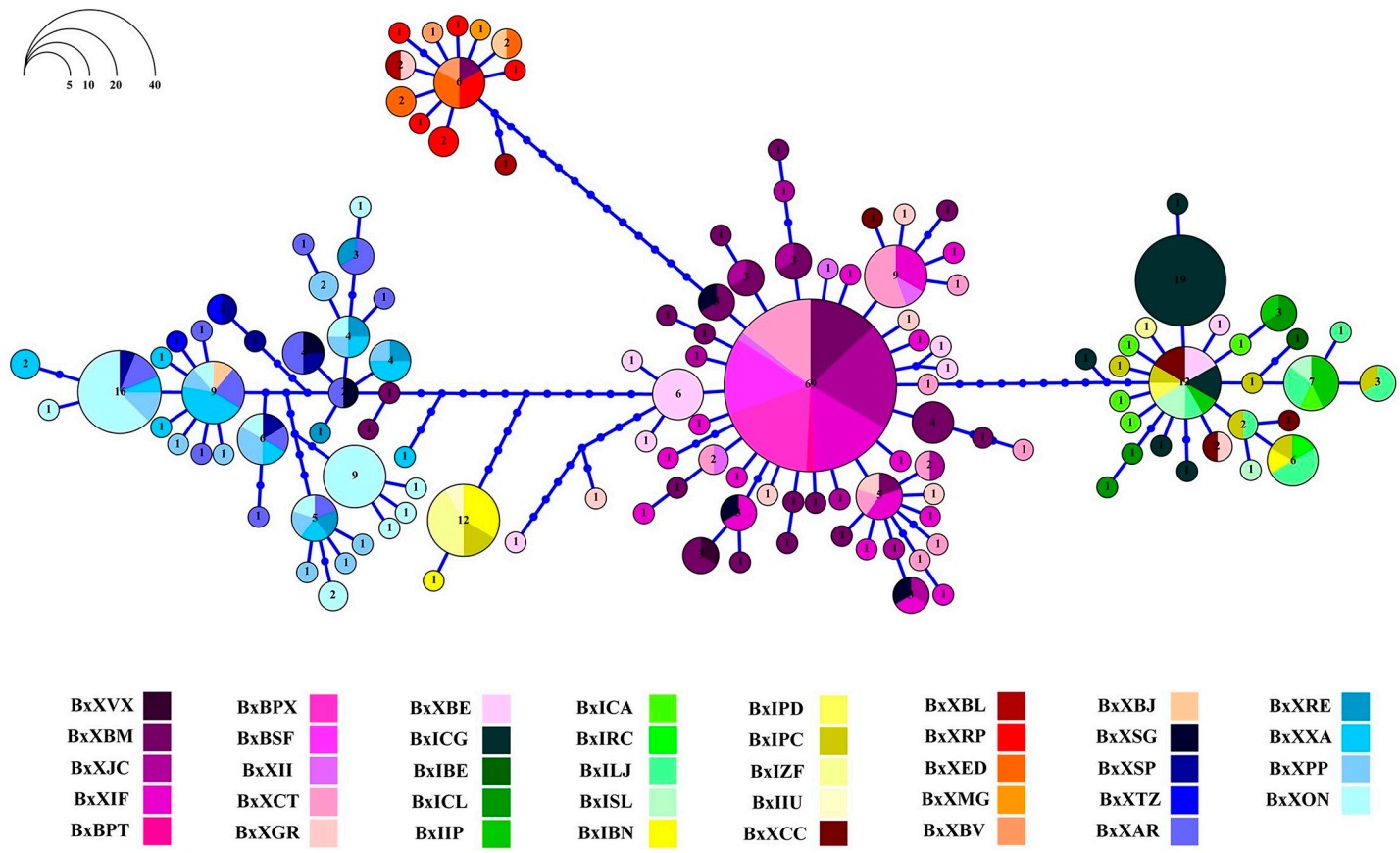

**Fig 4. Haplotype network showing the distribution of 134 haplotypes in *Baryancistrus xanthellus* based on the mitochondrial Cyt b marker.** Each color represents the 39 sampled sites in this study grouped accordingly: Iriri Resex (Group A–greenish), Xingu Resex (B–reddish), Volta Grande do Xingu (C–purplish), São Félix do Xingu (D–bluish), and Ecological Station of Terra do Meio (E–yellowish).

Resex (B) and Volta Grande do Xingu (C). Reduced $\pi$ values associated with high $h$ values usually indicate bottleneck events followed by population expansion, thereby leading to accumulation of new mutations [50, 51]. A similar pattern suggestive of population expansion was reported in another species of Loricariidae, *Hypostomus ancistroides*, [52], as well as in other families of Siluriformes [53–55].

On the contrary, the group from São Félix do Xingu (group D) in the upper Xingu presented both high $h$ and $\pi$ values, typical of large and stable populations over long periods of evolutionary history [51]. For the group from the Ecological Station of Terra do Meio (group E) in the upper Iriri, both $h$ and $\pi$ diversity values were relatively low. This result might be explained by the reduced sample size of *B. xanthellus* from this region, as also inferred for some lineages of other Siluriformes, such as *Bagre bagre* (Ariidae) [56].

When compared to the other groups, the lowest haplotype diversity was detected in the groups from Volta Grande do Xingu at the downstream limit of the distribution of *B. xanthellus*. The Volta Grande do Xingu hosts the largest rapids and bedrock channels within the Xingu basin. This particular physiography may provide higher habitat availability and continuity for populations of *B. xanthellus*, favoring gene flow. Moreover, Volta Grande do Xingu supports most of colour morphs observed in *B. xanthellus*, making this stretch popular among ornamental fishermen over the last 40 years. Another commercially important loricariid species, *Hypancistrus zebra*, is endemic to Volta Grande and classified as Critically Endangered by

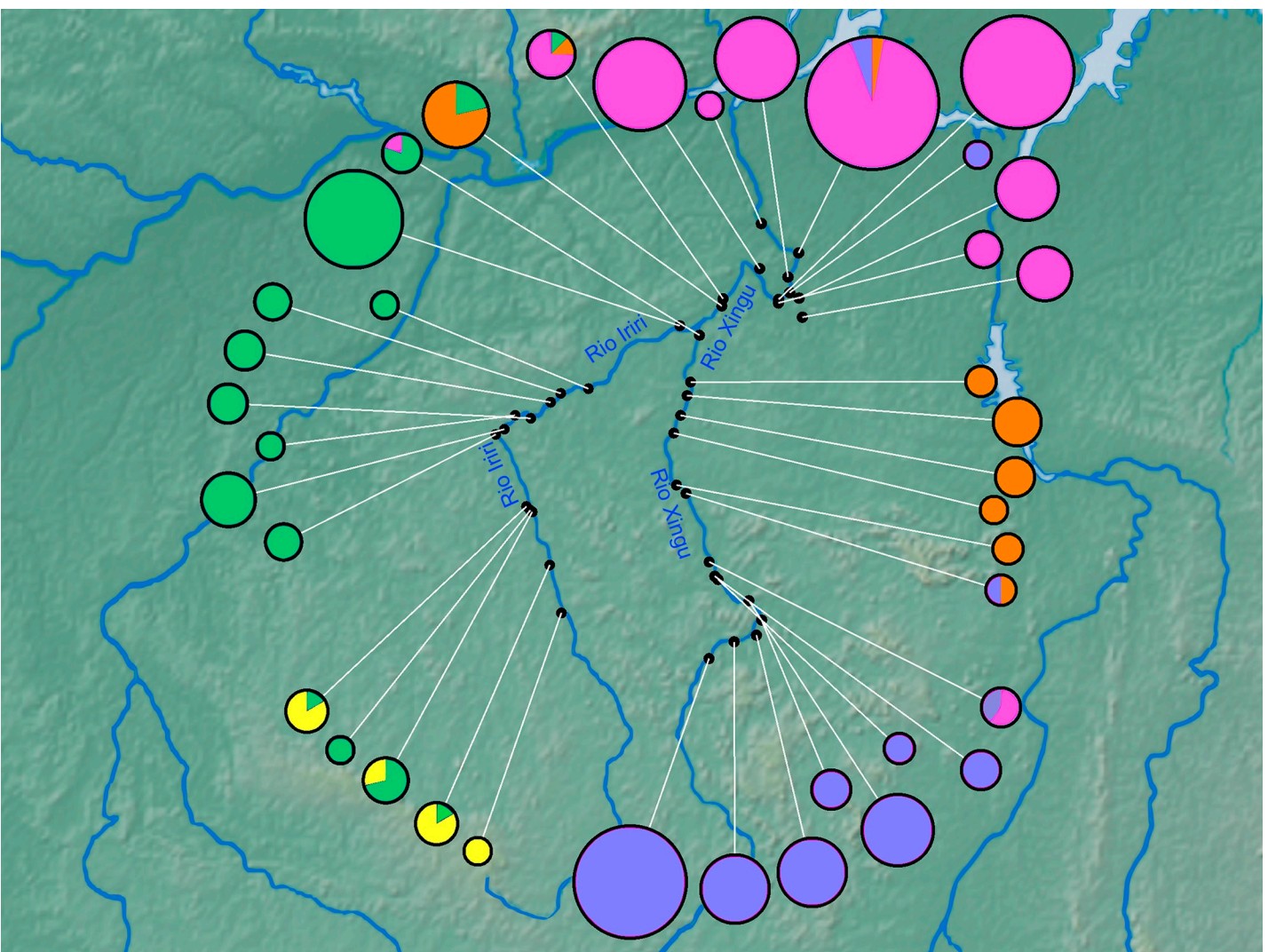

**Fig 5. Distribution of Cyt b haplotypes of *Baryancistrus xanthellus* assigned to Group A (Iriri Resex–green), Group B (Xingu Resex–orange); Group C (Volta Grande do Xingu–magenta); Group D (São Félix do Xingu–blue), and Group E (Ecological Station of Terra do Meio–yellow).**

the Brazilian Ministry of Environment [57] due to overfishing and habitat loss attributed to the Belo Monte dam complex. Accordingly, the group from this region also presented the highest number of shared haplotypes among individuals, suggesting the occurrence of population bottleneck.

The construction of the Belo Monte hydroelectric power plant, which began operations in the first quarter of 2016, might promote a spatial and temporal loss of genetic diversity in *B. xanthellus*. The population from Volta Grande do Xingu is directly affected by the loss of large rapids and availability of food resources due to sediment accumulation on rocky substrates in the reservoir and in the channel downstream of the impoundment dam [10, 11]. The change from lotic to lentic environments can cause drastic declines in populations of rheophilic and lithophilic fish species and eventually lead to local extinctions [24, 58].

The population structure inferred from Cyt b data revealed five haplogroups of *B. xanthellus* distributed along the Xingu River basin. These lineages were characterized by higher levels of intraspecific variation that indicate a strong population genetic substructure pattern.

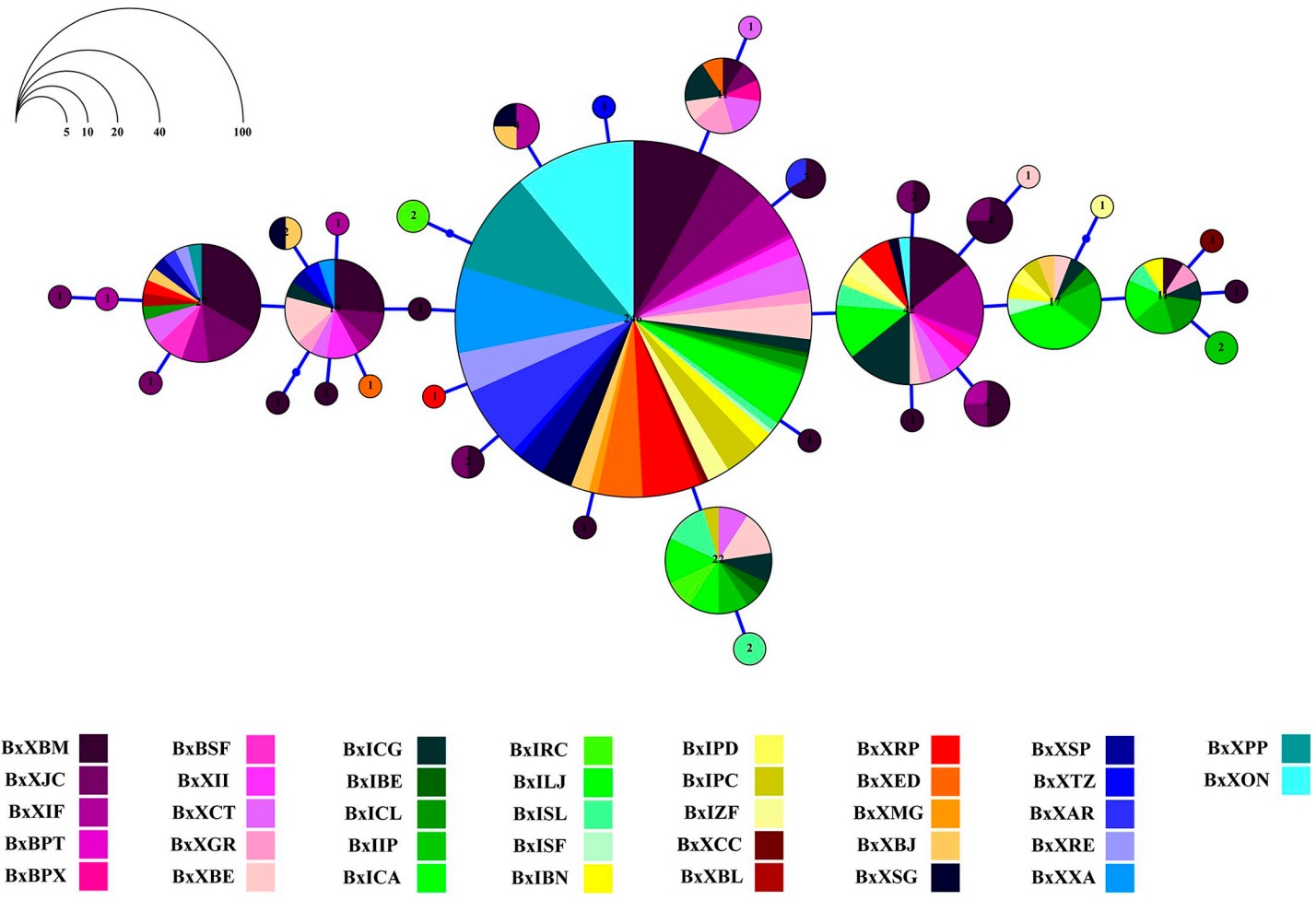

**Fig 6. Haplotype network showing the distribution of 60 haplotypes in *Baryancistrus xanthellus* based on the nuclear RPL3 marker.** Each color represents the 39 sampled sites in this study. The colors by group are: Iriri Resex (Group A–greenish), Xingu Resex (B–reddish), Volta Grande do Xingu (C–purplish), São Félix do Xingu (D–bluish), and Ecological Station of Terra do Meio (E–yellowish).

In fact, the Xingu and Iriri rivers encompass high variety of aquatic environments, including powerful rapids with rocky substrates interspersed with backwaters with bottoms of sand or gravel [59]. Both rivers are often split into multiple channels with variable depths and levels of connectivity, the latter including steep drops that form waterfalls. This extreme variation in hydrodynamics and riverbeds promotes the fragmentation of aquatic habitats in the Xingu River, where rocky rapids are separated by slackwater stretches with substrates dominated by sediments.

*Baryancistrus xanthellus* is highly adapted to rocky rapids and, like most loricariid species, is phylopatric with no significant migration over breeding periods. These features might restrain gene flow among populations along a single bedrock river stretch [8]. Considering the longitudinal distribution of the five haplogroups herein identified, our observations of large backwater areas with deep channels and sediment-laden bottoms between rocky rapids might contribute to the limited dispersal of local populations. In other words, the isolation of shallow rocky rapids by large stretches of not so complex river channel with more calm waters leading to flow expansion and sediment accumulation combined with the biological particularities of *B. xanthellus* likely influenced genetic differentiation among populations of this species.

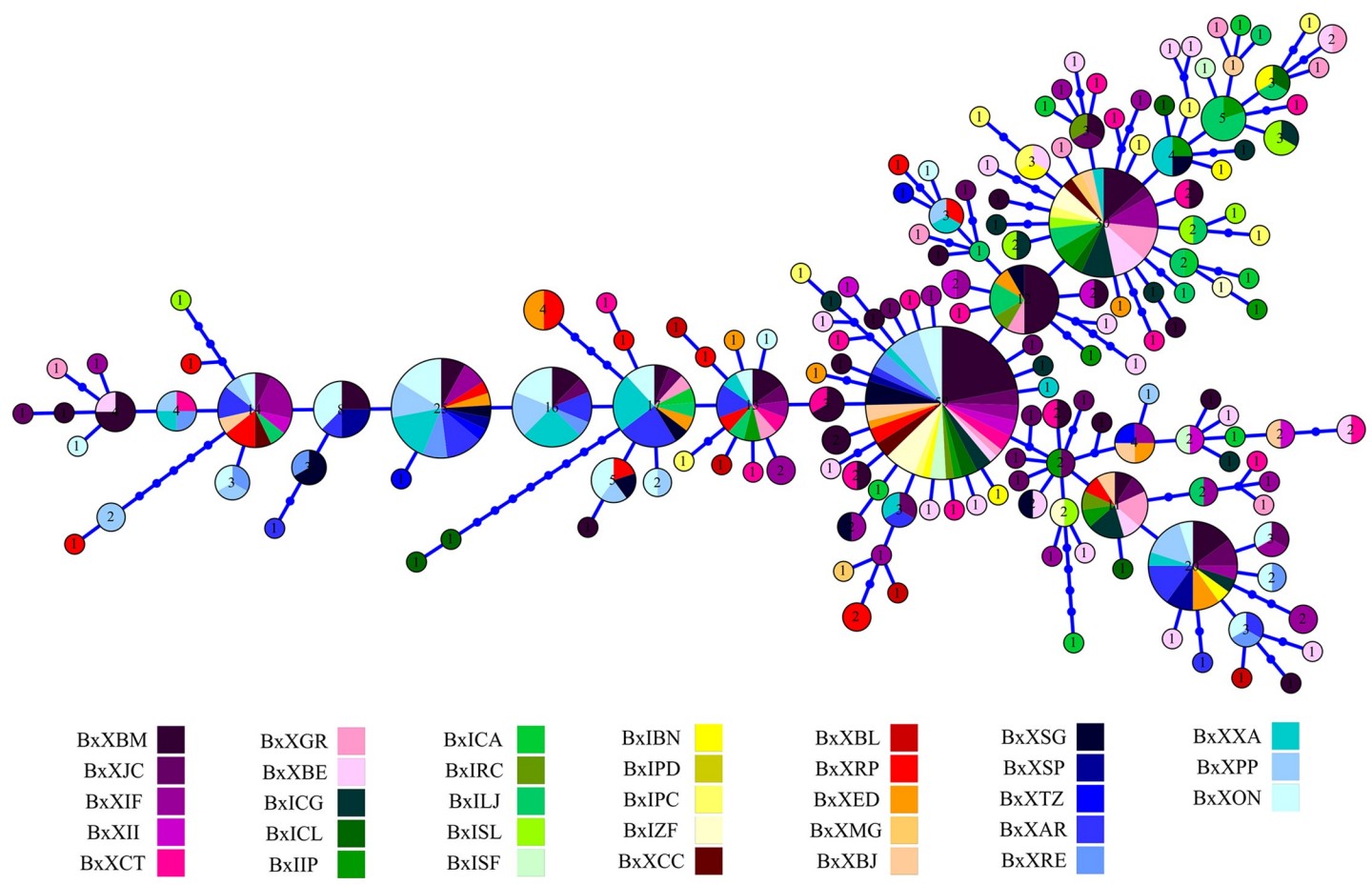

**Fig 7. Haplotype network showing the distribution of 156 haplotypes in *Baryancistrus xanthellus* based on the nuclear PRL marker.** Each color represents the sampled sites in this study. The colors by group are: Iriri Resex (Group A–greenish), Xingu Resex (B–reddish), Volta Grande do Xingu (C–purplish), São Félix do Xingu (D–bluish), and Ecological Station of Terra do Meio (E–yellowish).

Nonetheless, some haplotyes of the Cyt b marker were shared between both neighboring and more distant populations. Putatively, this pattern points to a dynamic environment, wherein sediment accumulation and erosion would drive the fragmentation and expansion of rocky substrates. The sediment accumulation within rivers across space and through time depends on the interaction between channel morphology and water discharge. In the local scale, zones of channel widening and flow expansion are susceptible to sediment accumulation [60]. This would drive the distribution of riverbeds with exposed rocks or covered by sediments under a specific hydrological regime. Zones susceptible for sediment accumulation are also formed by the backwater effect of the trunk river in its tributaries. In this case, rocky riverbeds would be more fragmented or less persistent in the downstream reach of tributaries, difficulting the contact between populations of the trunk river (Xingu) and its tributaries (Iriri and Bacajá). In a thousand years timescale, periods of higher precipitation in eastern Amazonia increase the water discharge of the Xingu and Iriri rivers, favoring lateral or vertical channel erosion, sediment bypass and the expansion of zones of rocky rapids. This condition would promote some degree of gene flow among populations of *B. xanthellus*. On the other hand, dry periods would induce fluvial aggradation through accumulation of sediment within river channels [61], increasing the fragmentation of rocky rapids.

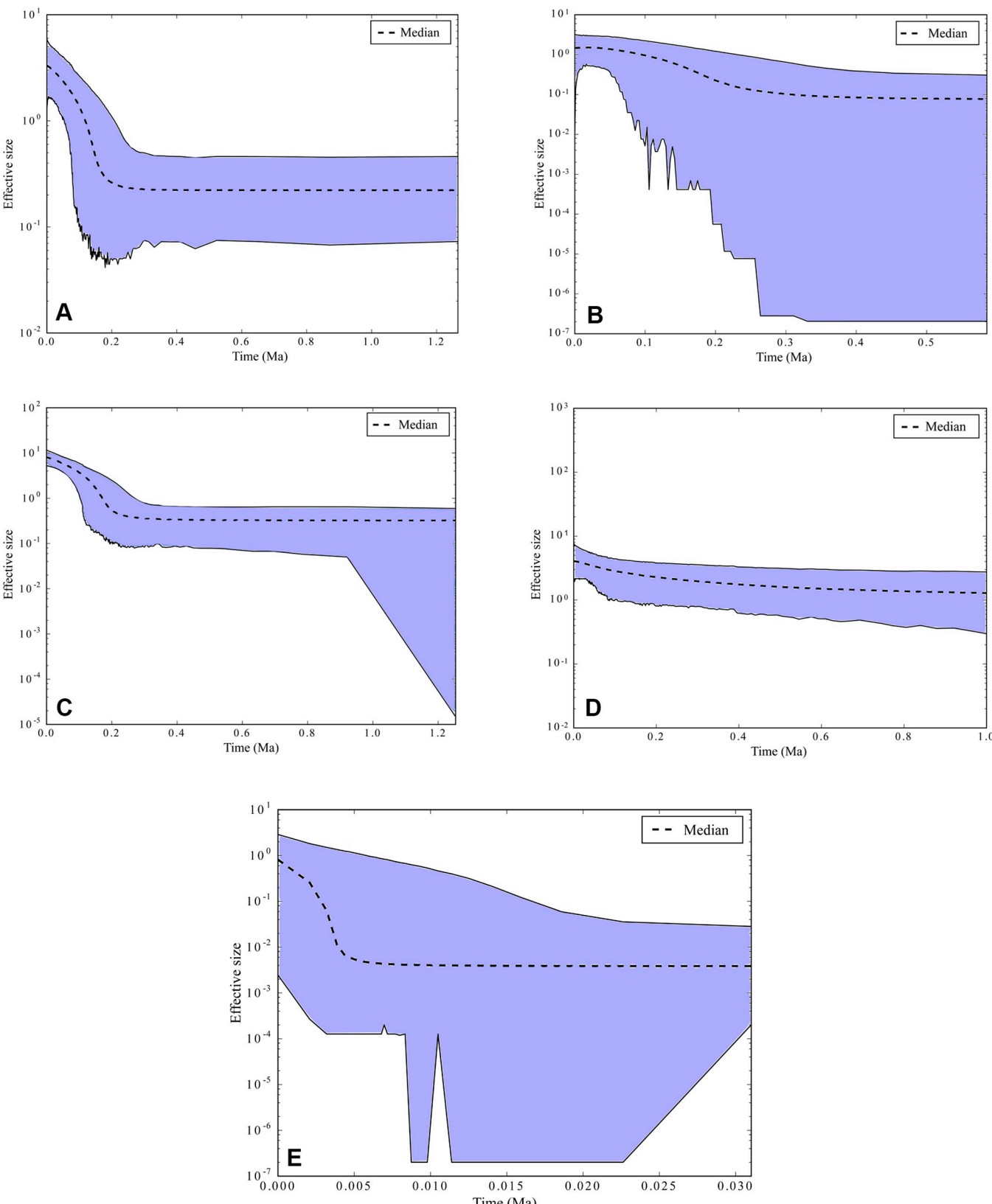

**Fig 8. Bayesian Skyline Plot for the five lineages (Groups A–E) of *Baryancistrus xanthellus* from the Xingu River basin.** The curves correspond to the concatenated markers. The y axis corresponds to the effective population size (Ne) and the x axis corresponds to the mean period estimated in million years (Ma).

The rainfall in the Amazon region is driven by the South American monsoon system (SAMS) [62]. Eastern Amazonia, including the Xingu River basin, experienced remarkable variation in rainfall during the late Quaternary [63, 64]. For example, rainfall in eastern Amazonia was reduced during the last glacial maximum (~21–23 ka); conversely, rainfall may have increased during the middle Holocene (~6 ka) [64]. Abrupt climatic changes, such as the Heinrich Stadial 1 (HS1; ~18–15 ka), also promoted dramatic changes in rainfall and the water discharge of Amazonian rivers [65, 66].

Hydrology significantly impacts sediment transport and accumulation along river channels and consequently promotes changes in the continuity (or fragmentation) of rocky substrates. For eastern Amazonian rivers, periods of higher rainfall during the middle Holocene and HS1, would have increased the connectivity of fluvial channels and rocky rapids. On the other hand, the shift to a drier climate during last glacial maximum would have favoured the accumulation of sediments and the fragmentation of rocky rapids, isolating populations of species adapted to these habitats, such as *B. xanthellus*. Besides the climate variations reported since the last glacial maximum, the rainfall across Amazonia was modulated by insolation cycles with ~25 ka periodicity during the last 250 ka [63, 67]. These variations in hydrology and their effect on the sedimentary budget imply dynamic long-term distribution of rocky and sediment substrates of the Xingu River.

Similar patterns of population structure have been reported in other loricariid catfishes, such as *Hypostomus*. Based on the mitochondrial ATPase 6/8 marker, Hollanda et al. [52] identified 12 highly structured and genetically divergent populations of *H. ancistroides* from Paraná River basin. According to these authors, this pattern was related to the biological features of the species and to changes in environmental conditions. Also using ATPase 6/8 sequences, Borba et al. [68] performed phylogeographic and phylogenetic analyses on populations of *Hypostomus strigaticeps* from four sub-basins of the Paraná River. Their results confirmed that populations could be discriminated into two cryptic lineages within the single species.

Unlike those based on mtDNA, analyses based on nuclear markers often fail to detect subdivisions between populations, as was the case for *B. xanthellus*. Nuclear markers have lower rates of evolution when compared to mitochondrial markers, and are therefore less sensitive to recent events of population division [69].

## Demographic history

Results obtained for nearly all samples by both *D* and *Fs* statistic tests suggest deviation of neutrality. Negative values may be interpreted as population expansion as suggested by the haplotype network star-like shape and by EBSP results [44]. In addition, the EBSP curves corroborated the neutrality tests, suggesting that population expansions in the A and C groups initiated about 200 ka. In contrast, the B and D groups seem remained stable over this period.

As previously discussed, rainfall in eastern Amazonia combined with channel hydrodynamics (zones of flow expansion and confluences) drives the capacity of clearwater rivers to trap (low rainfall) or transport (high rainfall) sediments to the Amazon River channel. Shield areas have very low rates of surface erosion [70], which point to relatively stable relief in the thousand years timescale. Thus, historical changes in rainfall and their consequent effect on fluvial sedimentation thereby have higher impact to the size and connectivity of rocky rapids during the late Quaternary.

Population expansion in groups A and C of *B. xanthellus* estimated over the last 200 Kya points to an increased availability of rocky rapids in the Iriri River and downstream stretch of the Xingu River. On the other hand, population stability in groups B and D suggests that aquatic environments in the Xingu River responded differently to regional climate changes, possibly due to hydrodynamic controls on the spatial distribution of sediment accumulation zones. These environmental dynamics in response to changes in rainfall could account for the high genetic diversity and population expansion or stability of *B. xanthellus* from different portions of the Xingu River basin. This hypothesis implies that similar phylogeographic patterns should be present in other loricariid catfishes sharing equivalent biological features and generation intervals.

## Final remarks

This is the first study to characterize the phylogeographic pattern in *B. xanthellus* using mitochondrial and nuclear markers and encompassing a wide sampling over the entire species geographic range. Analyses of the Cyt b marker inferred five longitudinally distributed haplogroups with high genetic diversity in *B. xanthellus*. Those haplogroups are putatively associated with the philopatric nature of this species and the historical isolation of its preferred habitat (rocky rapids) driven by fluvial hydrodynamics and hydrological changes in eastern Amazonia over the late Quaternary. Analyses of the nuclear markers did not reveal a similar pattern due the recent nature of such diversification.

The demographic inferences indicated that some haplogroups have undergone population expansion while others have experienced population stability, both probably associated with spatial variation in aquatic habitats due to late Quaternary climate changes. Changes in water discharge driven by long-term changes in rainfall shifted the riverine landscape via sediment transport or accumulation in the main channels. Consequently, rocky rapids have expanded (via sediment bypass) or retracted and fragmented (via sediment accumulation) along different stretches of the Xingu and Iriri Rivers during the last 200 ka.

Conservation strategies for *B. xanthellus* must take this phylogeographic pattern into account. The building and operation of the Belo Monte hydropower plant have significantly impacted the availability and connectivity of rocky rapids in the Volta Grande stretch of the Xingu channel and will likely reduce levels of genetic diversity and gene flow among resident populations. Furthermore, the ornamental fish trade of loricariids such as *B. xanthellus* should be monitored to avoid overfishing at sensitive localities.

## Acknowledgments

We thank the curators and staff of the Laboratório de Ictiologia de Altamira, (UFPA/Altamira), Laboratório de Genética Aplicada, and Laboratório de Ictiologia e Biodiversidade Subterrânea (UFPA/IECOS-Bragança) for providing samples. Thanks to Marcella Santos and Mariangeles Arce for their important comments on earlier drafts of the manuscript. We thank Instituto Chico Mendes de Conservação da Biodiversidade (ICMBio), which through the ARPA consortium provided logistic and support for fieldwork.

## Author Contributions

**Conceptualization:** Keila Xavier Magalhães, André Oliveira Sawakuchi, Alany Pedrosa Gonçalves, Grazielle Fernanda Evangelista Gomes, Janice Muriel-Cunha, Leandro Melo de Sousa.

**Data curation:** Keila Xavier Magalhães, Raimundo Darley Figueiredo da Silva, Grazielle Fernanda Evangelista Gomes, Janice Muriel-Cunha, Leandro Melo de Sousa.

**Formal analysis:** Keila Xavier Magalhães, Raimundo Darley Figueiredo da Silva, André Oliveira Sawakuchi, Grazielle Fernanda Evangelista Gomes, Janice Muriel-Cunha.

**Funding acquisition:** André Oliveira Sawakuchi, Grazielle Fernanda Evangelista Gomes, Janice Muriel-Cunha, Mark H. Sabaj, Leandro Melo de Sousa.

**Investigation:** Keila Xavier Magalhães, Raimundo Darley Figueiredo da Silva, André Oliveira Sawakuchi, Alany Pedrosa Gonçalves, Grazielle Fernanda Evangelista Gomes, Janice Muriel-Cunha, Leandro Melo de Sousa.

**Methodology:** Keila Xavier Magalhães, André Oliveira Sawakuchi, Alany Pedrosa Gonçalves, Grazielle Fernanda Evangelista Gomes, Janice Muriel-Cunha, Leandro Melo de Sousa.

**Project administration:** Leandro Melo de Sousa.

**Resources:** André Oliveira Sawakuchi, Grazielle Fernanda Evangelista Gomes, Mark H. Sabaj, Leandro Melo de Sousa.

**Software:** Raimundo Darley Figueiredo da Silva.

**Supervision:** Grazielle Fernanda Evangelista Gomes, Janice Muriel-Cunha, Leandro Melo de Sousa.

**Validation:** André Oliveira Sawakuchi, Grazielle Fernanda Evangelista Gomes, Janice Muriel-Cunha, Mark H. Sabaj, Leandro Melo de Sousa.

**Visualization:** Grazielle Fernanda Evangelista Gomes, Mark H. Sabaj.

**Writing – original draft:** Keila Xavier Magalhães, Raimundo Darley Figueiredo da Silva, André Oliveira Sawakuchi, Alany Pedrosa Gonçalves, Grazielle Fernanda Evangelista Gomes, Janice Muriel-Cunha, Mark H. Sabaj, Leandro Melo de Sousa.

**Writing – review & editing:** Keila Xavier Magalhães, Raimundo Darley Figueiredo da Silva, André Oliveira Sawakuchi, Alany Pedrosa Gonçalves, Grazielle Fernanda Evangelista Gomes, Janice Muriel-Cunha, Mark H. Sabaj, Leandro Melo de Sousa.

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
