## [Decision Letter · Decision Letter 0]

16 Jun 2021

PONE-D-21-10171

Phylogeography of Baryancistrus xanthellus (Siluriformes: Loricariidae), a rheophilic catfish endemic to the Xingu River basin in eastern Amazonia

PLOS ONE

Dear Dr. Sousa,

Thank you for submitting your manuscript to PLOS ONE. After careful consideration, we feel that it has merit but does not fully meet PLOS ONE’s publication criteria as it currently stands. Therefore, we invite you to submit a revised version of the manuscript that addresses the points raised during the review process.

Both reviewers had relatively few comments and suggestions and they agree that the manuscipt should be published in this journal. 

We look forward to receiving your revised manuscript.

Kind regards,

Juan Marcos Mirande

Academic Editor

PLOS ONE

Journal Requirements:

Reviewers' comments:

Reviewer's Responses to Questions

**Comments to the Author**

1. Is the manuscript technically sound, and do the data support the conclusions?

Reviewer #1: Yes

Reviewer #2: Yes

2. Has the statistical analysis been performed appropriately and rigorously? 

Reviewer #1: Yes

Reviewer #2: N/A

3. Have the authors made all data underlying the findings in their manuscript fully available?

Reviewer #1: Yes

Reviewer #2: Yes

4. Is the manuscript presented in an intelligible fashion and written in standard English?

Reviewer #1: Yes

Reviewer #2: Yes

5. Review Comments to the Author

Reviewer #1: The manuscript deals with a variable catfish species in a proper way, analyzing genetically different subpopulations and concluding about their phylogeographic history. This study was long needed and is presently beautifully. The manuscript is generally well researched and well written, and include very good, highly illustrative figures.

I only have a few corrections before acceptance, which are listed below, by line number:

Lines 7, 9, 10, 12 and 14: Please add the postal code (CEP) to all Brazilian addresses before the city name, and without a separating comma.

Line 51: It is numerous but obviously countable… so countless is not a good adjective. Replace by numerous or other similar word.]

Line 57: Those three basins .. drain the …

Line 58: ... their downstream portions lie on the …

61: Xingu River

81: color (stick to American English)

104: middle Xingu River (middle is not part of the name)

109: Xingu River basin (basin is not part of the name)

117 and 246: I think “ria” is not an English word. It is usually translated as estuary, despite a “ria” opens to another river. Perhaps you should explain the meaning of “ria” at first usage in M&M, or chose a different way of saying ria.

119: middle Xingu

314: Demographic

347: xanthellus

351: Critically Endangered, without quotes and with capital initials (as recommended by IUCN).

419: add space after rapids,

454: responded

463-4: over the entire species geographic range.

Congratulations on the very good figures!

R. E. Reis

Reviewer #2: Review Comments to the Author

Please use the space provided to explain your answers to the questions above. You may also include additional comments for the author, including concerns about dual publication, research ethics, or publication ethics. (Please upload your review as an attachment if it exceeds 20,000 characters) (Limit 200 to 20000 Characters)

Could you please see my attached review, I have added all my comments to the Author in an attachment.

6. PLOS authors have the option to publish the peer review history of their article (what does this mean?). If published, this will include your full peer review and any attached files.

Reviewer #1: No

Reviewer #2: No

---

## [Author Response · Author response to Decision Letter 0]

27 Jul 2021

COMMENTS

Comments from Reviewer 1

The manuscript deals with a variable catfish species in a proper way, analyzing genetically different subpopulations and concluding about their phylogeographic history. This study was long needed and is presently beautifully. The manuscript is generally well researched and well written, and include very good, highly illustrative figures.

I only have a few corrections before acceptance, which are listed below, by line number:

Response: We have incorporated the suggestions throughout the manuscript.

• Comment: Lines 7, 9, 10, 12 and 14: Please add the postal code (CEP) to all Brazilian addresses before the city name, and without a separating comma.

Response: Done. 

• Comment: Line 51: It is numerous but obviously countable… so countless is not a good adjective. Replace by numerous or other similar word.]

Response: Done. 

• Comment: Line 57: Those three basins .. drain the …

Response: Done. 

• Comment: Line 58: ... their downstream portions lie on the …

Response: Done. 

• Comment: Line 61: Xingu River

Response: Done. 

• Comment: Line 81: color (stick to American English).

Response: Done.

• Comment: Line 104: middle Xingu River (middle is not part of the name).

Response: Done. 

• Comment: Line 109: Xingu River basin (basin is not part of the name).

Response: Done. 

• Comment: Line 117 and 246: I think “ria” is not an English word. It is usually translated as estuary, despite a “ria” opens to another river. Perhaps you should explain the meaning of “ria” at first usage in M&M or chose a different way of saying ria.

Response: The term is explained in Introduction, line 61.

• Comment: Line 119: middle Xingu.

Response: Done. 

• Comment: Line 314: Demographic.

Response: Done. 

• Comment: Line 347: xanthellus.

Response: Done. 

• Comment: Line 351: Critically Endangered, without quotes and with capital initials (as recommended by IUCN).

Response: Done.

• Comment: Line 419: add space after rapids,

Response: Done. 

• Comment: Line 454: responded.

Response: Done. 

• Comment: Line 463-4: over the entire species geographic range.

Response: Done. 

Comments from Reviewer 2

• Comment: 164 - 173: I’m not familiar with Phase algorithms but I know that it offers a number of parameters. It is important that these parameters are given on the text.

Response: Done. In the new version of the manuscript a sentence about PHASE parameters is included (lines 185-190).

• Comment: Parameters of BAPS6 are also poorly described. I strongly recommend the authors to have a look at other papers to see how methods are explained. Number of replications and K values are fundamental and need to be explained. How did K was estimated? Was it based on the haplotype network results? 

Response: Done. In the new version of the manuscript BAPS section was improved in material and methods (lines 201-206).

• Comment: Did authors use other software to build the output like Distruct or Clumpp? 

Response: BAPS does not require the use of extra packages for graphically displays results of runs.

• Comment: Finally, I never used BAPS and I was unable to install or see its documents and for these reasons I cannot assure what are the parameters that should figure on the text. Though I believe it might be close to Structure package. If this is the case, authors should keep in mind that this is an analysis of attribution that is dependent of K calculation, where K is the number of populations. Different numbers of K should have been compared in order to find the best population structure given the current dataset. So, its of paramount importance to let the reader know how K was found.

Response: We tested different levels of K (1-20). The number of K was chosen based on highest posterior probability value. See BAPS section in material and methods and Results/Population genetic structure section (lines 201-206).

• Comment: 185 - 192: AMOVA was employed in a manner that I do not agree and consequently its results are not worthy to be discussed. In fact, AMOVA is usually used in two ways. The first as an exploratory analysis to help the formulation of hypothesis given a data set. The second is to test and compare different hypotheses, i.e. different scenarios of population structure. In this work, an analysis of attribution is performed by BAPS6 and its result is considered to be the most plausible population structure. Hence, AMOVA should be used to go further on BAPS proposed structure and to show in more detail how genetic variation is partitioned among populations. I also believe that FST was used as an exploratory analysis and its results are not relevant. Since FST is also calculated in AMOVA I believe that this analysis could be removed from the paper.

Response: Done. We employed AMOVA based on BAPS structure. Text explained was added (lines 207-210). We also removed FST.

• Comment: 196: Parameters and priors of EBSP should also be stated.

Response: Done. Lines 217-219.

• Comment: 228: Table 3 caption should state what are one or two asterisks.

Response: Done.

• Comment: 265-267: This is not correct. Results for cytb are distinctly different.

Response: Corrected. We inserted the result of AMOVA based on BAPS structure.

• Comment: 276: This whole table of results is unnecessary (as stated above - see comments on the use of FST and AMOVA).

Response: We decided to keep this table as we are still using AMOVA (but removed FST, as explained above).

• Comment: 279: AMOVA, and not FST, should have been used to test the hypothesis of existing five population hypotheses (as stated above - see comments on the use of FST and AMOVA)

Response: Addressed above.

• Comment: 301: “The lack of genetic substructure ‘in’ nuclear markers...” Is this sentence correctly written?

Response: Yes.

• Comment: Genetic diversity: Maybe this subsection could be merged with Population Structure and Phylogeography as in the very beginning it refers to the main result of population structure (“...for three of the five groups analyzed...”). This is just a suggestion. Demographic history.

Response: Done. 

• Comment: 439-441: Results of D and Fs can be interpreted further than only deviation from neutrality. Negative values may be interpreted as population expansion as suggested by the haplotype network star-like shape and endorsed by EBSP results.

Response: Done. See Demographic history section on Discussion.

---

## [Editor Report · Decision Letter 1]

10 Aug 2021

PONE-D-21-10171R1

Phylogeography of *Baryancistrus xanthellus* (Siluriformes: Loricariidae), a rheophilic catfish endemic to the Xingu River basin in eastern Amazonia

PLOS ONE

Dear Dr. Sousa,

All the suggestions from the reviewers were followed and I find the manuscript ready to be published.

I just found one word that went back to British English in the latest review, that should be kept in American English to be congruent. In Line 114, please change back "Analysed" to "Analyzed". As the journal hasn't copy-editing, it should be done by the authors.

After that, the manuscript will be sent to production.

Sincerely, Marcos Mirande
---

## [Author Response · Author response to Decision Letter 1]

11 Aug 2021

As asked in the latest Decision Letter (copied below), the word "Analysed" was changed to "Analyzed".

Dear Dr. Sousa,

All the suggestions from the reviewers were followed and I find the manuscript ready to be published.

I just found one word that went back to British English in the latest review, that should be kept in American English to be congruent. In Line 114, please change back "Analysed" to "Analyzed". As the journal hasn't copy-editing, it should be done by the authors.

After that, the manuscript will be sent to production.

---

## [Editor Report · Decision Letter 2]

13 Aug 2021

Phylogeography of *Baryancistrus xanthellus* (Siluriformes: Loricariidae), a rheophilic catfish endemic to the Xingu River basin in eastern Amazonia

PONE-D-21-10171R2

Dear Dr. Sousa,

We’re pleased to inform you that your manuscript has been judged scientifically suitable for publication and will be formally accepted for publication once it meets all outstanding technical requirements.

Kind regards,

Juan Marcos Mirande

Academic Editor

PLOS ONE
---

## [Editor Report · Acceptance letter]

20 Aug 2021

PONE-D-21-10171R2 

Phylogeography of *Baryancistrus xanthellus* (Siluriformes: Loricariidae), a rheophilic catfish endemic to the Xingu River basin in eastern Amazonia 

Dear Dr. Sousa:

I'm pleased to inform you that your manuscript has been deemed suitable for publication in PLOS ONE. Congratulations! Your manuscript is now with our production department. 

Kind regards, 

on behalf of

Dr. Juan Marcos Mirande 

Academic Editor

PLOS ONE